# Exploring Macrocyclic Chemical Space: Strategies and Technologies for Drug Discovery

**DOI:** 10.3390/ph18050617

**Published:** 2025-04-24

**Authors:** Taegwan Kim, Eunbee Baek, Jonghoon Kim

**Affiliations:** 1Department of Chemistry, Integrative Institute of Basic Science, Soongsil University, Seoul 06978, Republic of Korea; lever19787@soongsil.ac.kr; 2Department of Green Chemistry and Materials Engineering, Soongsil University, Seoul 06978, Republic of Korea; eunbi112@soongsil.ac.kr

**Keywords:** macrocyclic ring synthesis, molecular diversity, natural product mimics drug discovery, computational design

## Abstract

Macrocycles have emerged as significant therapeutic candidates in drug discovery due to their unique capacity to target complex and traditionally inaccessible biological interfaces. Their structurally constrained three-dimensional configurations facilitate high-affinity interactions with challenging targets, notably protein–protein interfaces. However, despite their potential, the synthesis and optimization of macrocyclic compounds present considerable challenges related to structural complexity, synthetic accessibility, and the attainment of favorable drug-like properties, particularly cell permeability and oral bioavailability. Recent advancements in synthetic methodologies have expanded the chemical space accessible to macrocycles, enabling the creation of structurally diverse and pharmacologically active compounds. Concurrent developments in computational strategies have further enhanced macrocycle design, providing valuable insights into structural optimization and predicting molecular properties essential for therapeutic efficacy. Additionally, a deeper understanding of macrocycles’ conformational adaptability, especially their ability to internally shield polar functionalities to improve membrane permeability, has significantly informed their rational design. This review discusses recent innovations in synthetic and computational methodologies that have advanced macrocycle drug discovery over the past five years. It emphasizes the importance of integrating these strategies to overcome existing challenges, illustrating how their synergy expands the therapeutic potential and chemical diversity of macrocycles. Selected case studies underscore the practical impact of these integrated approaches, highlighting promising therapeutic applications across diverse biomedical targets.

## 1. Introduction

Natural products have long been recognized as a rich and invaluable source of bioactive compounds, offering an extensive array of structurally diverse and complex molecules that have evolved over millions of years to finely interact with biological systems [1,2,3]. These small molecules, derived from diverse organisms such as plants, microorganisms, and marine life, frequently possess intricate three-dimensional architectures and unique chemical features that are challenging to replicate through synthetic approaches alone [4,5,6,7,8]. This inherent structural complexity and diversity enable natural products to interact with multiple biological targets simultaneously, often leading to enhanced efficacy and novel mechanisms of action [9,10,11,12].

The exceptional properties of natural product-derived small molecules render them particularly valuable in drug discovery. Their ability to modulate biological processes in ways that purely synthetic compounds often cannot has facilitated the development of numerous successful pharmaceuticals across a diverse array of therapeutic domains, including antibiotics [13,14,15,16], anticancer agents [17,18,19], and immunosuppressants [20,21]. Furthermore, natural products frequently serve as ideal starting points for medicinal chemistry efforts, enabling researchers to optimize their pharmacological properties and generate semi-synthetic derivatives with improved potency, selectivity, and pharmacokinetic profiles [10,22,23,24,25,26,27]. As the discipline of drug discovery progresses, the ongoing investigation of natural products remains an essential approach for identifying novel lead compounds and broadening the available chemical space for therapeutic development.

Macrocycles, typically characterized as cyclic structures comprising 12 or more atoms, are distinguished by their intricate three-dimensional architectures and distinctive chemical properties. These attributes enable macrocycles to interact with biological targets that are often inaccessible to conventional small molecules [28,29,30,31,32,33]. In particular, macrocyclic natural products and their analogues have emerged as powerful therapeutic candidates against otherwise “undruggable” targets, owing to their constrained 3D conformations that pre-organize the molecule (lowering the entropic penalty upon binding) and allow high-affinity interactions across extended protein surfaces [34,35,36,37]. This preorganization means macrocycles can form extensive contacts with shallow or flat binding sites (e.g., protein–protein interaction interfaces), often yielding enhanced binding affinity, selectivity, and even pharmacological properties relative to linear analogues. Indeed, macrocyclic scaffolds bridge the gap between traditional small molecules and larger biologics, exemplified by the fact that macrocycles dominate the inhibitors of certain challenging targets (for instance, hepatitis C virus protease NS3/4A, which has a solvent-exposed shallow groove) [38].

These attributes have led to numerous successful macrocyclic drugs (e.g., antibiotics, anticancer agents, immunosuppressants), and as of recent counts over 80 macrocyclic drugs have been approved for clinical use (Table 1) [39]. In addition to these established agents, a growing number of macrocyclic compounds are currently undergoing clinical and preclinical evaluation across various therapeutic domains, including oncology and infectious diseases (Table 2). This trend underscores the expanding significance of macrocycles in contemporary drug discovery. Despite their promise, the therapeutic exploitation of macrocycles is hindered by difficulties in chemical synthesis and optimization. Natural macrocycles often contain multiple stereocenters [40,41,42,43] and sensitive functional groups [29,44,45], making them challenging to produce and modify. Specifically, macrocyclization reactions frequently suffer from inherent entropic penalties and competing oligomerization, typically requiring high-dilution conditions that reduce yields and scalability [46,47,48,49,50].

Consequently, the generation of extensive libraries of macrocycles for screening purposes, such as in structure–activity relationship studies, presents significant challenges. As a result, numerous potentially valuable macrocyclic scaffolds remain insufficiently explored in the context of drug discovery. To address these challenges, several significant strategies have been developed to broaden the accessible chemical space of macrocycles. The Build/Couple/Pair (B/C/P) combinatorial approach, as introduced by Spring et al., has facilitated the development of structurally diverse macrocycle libraries [60,61,62]. These combinatorial methods enable the generation of macrocycles that exhibit either extensive scaffold diversity or libraries oriented towards biologically relevant motifs (Figure 1a) [63,64]. Furthermore, the Complexity-to-Diversity (CtD) concept utilizes complex natural product frameworks as starting points, systematically deriving novel macrocyclic variants that enhance chemical diversity while maintaining bioactivity (Figure 1b) [65]. Despite these advances, the efficient and scalable synthesis of macrocyclic rings remains a significant and ongoing challenge.

**Table 2 pharmaceuticals-18-00617-t002:** Examples of recent developments in macrocyclic drug discovery.

Structure	Treatment	Activity	Reference
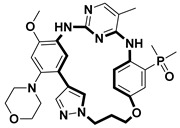	Anticancer	EGFR^T790M/C797S^ inhibitor	[66]
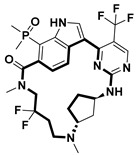	Anticancer	CDK7 inhibitor	[67]
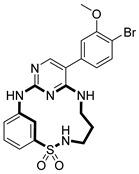	Prevention of dengue virus infection	EPHA2/A4 and GAK inhibitor	[68]
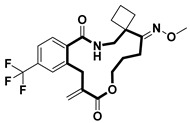	Virus inhibitor	H1N1 inhibitor	[69]
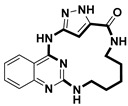	Anticancer	MST3 inhibitor	[70]
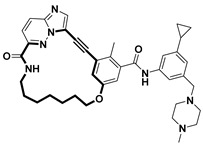	Anticancer	TRK inhibitor	[71]
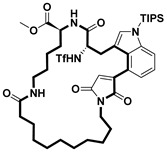	Virus inhibitor	SARS-CoV-2 inhibitor	[72]
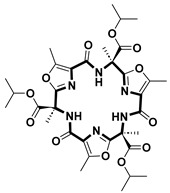	Virus inhibitor	SARS-CoV-2-main protease inhibitor	[73]
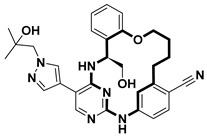	Anticancer	HPK1 inhibitor	[74]

In parallel to these synthetic efforts, computational methodologies have emerged as potent complementary strategies, significantly advancing macrocycle discovery. These approaches employ sophisticated algorithms and machine learning techniques to predict and optimize macrocycle structures, properties, and interactions with biological targets [75,76,77,78,79]. High-throughput virtual screening, molecular dynamics simulations, and structure-based design enable researchers to rapidly evaluate millions of macrocycle candidates in silico, thereby substantially reducing the time and resources required for experimental synthesis and testing. The integration of artificial intelligence (AI), particularly deep learning models, has further expedited macrocycle design by facilitating the identification of complex structure–activity relationships and the generation of novel scaffolds [28,80,81,82,83]. These AI and computational-driven strategies can utilize existing databases and experimental data to propose novel macrocycle structures with enhanced pharmacological properties, such as improved target selectivity and oral bioavailability [78,84,85,86]. The integration of computational methods with experimental validation has facilitated iterative design–test–optimize cycles, thereby rapidly refining macrocycle candidates and increasing discovery success rates.

In this review, we highlight these and other recent advances from the past five years, focusing specifically on innovative synthetic strategies—such as modular biomimetic assemblies [87,88], DNA-encoded libraries (DELs) [89]—as well as sophisticated computational design approaches [90,91,92]. In addition, this study examines the biological implications and therapeutic applications of macrocycles, focusing on essential factors such as cell permeability and bioavailability. Through selected case studies, we aim to illustrate how the synergy between innovative synthetic methods and powerful computational tools is expanding the macrocyclic chemical landscape, unlocking new therapeutic opportunities at the intersection of synthetic chemistry and drug discovery.

## 2. Modular Biomimetic Assembly Strategies for the Synthesis of Pseudonatural Macrocyclic Compounds

Recent advancements in synthetic methodologies have effectively addressed longstanding challenges in the construction of macrocycles. A notable instance is the modular biomimetic assembly approach developed by Yang and his team, which represents a significant advancement in the synthesis of structurally diverse macrocyclic molecules that mimic natural products [93]. This method holds considerable promise in the fields of medicinal chemistry and drug discovery. It effectively exploits the inherent biological activity of natural product scaffolds, while providing increased synthetic flexibility and an expanded chemical diversity beyond the limits of traditional natural products. A key aspect of this strategy is the retention of essential structural elements found in natural products, including privileged scaffolds, macrocyclic ring systems, specific stereochemical configurations, and critical functional groups responsible for targeted biological interactions. Traditional macrocyclization methods, such as peptide coupling [94,95], click chemistry [96,97,98], or ring-closing metathesis [99,100], often encounter limitations in scope and efficiency due to entropic challenges associated with large-ring formation [46,49]. However, the modular biomimetic assembly strategy overcomes these obstacles by introducing novel macrocyclization reactions that are more efficient and broadly applicable. This approach not only preserves the potent bioactivities inherent in natural compounds but also enables the exploration of previously inaccessible macrocyclic chemical space. This section highlights recent advancements, emphasizing the versatility, efficiency, and practical applications of this promising biomimetic modularization strategy through representative examples from the literature.

A representative example illustrating the power of modular biomimetic assembly strategies is the recent advancement combining biomimetic design with transition-metal catalysis for constructing structurally complex macrocycles. For instance, a recent innovative method inspired by enzymatic oxidation mechanisms employs rhodium(III)-catalyzed dual C–H/O₂ activation, enabling unprecedented macrocyclization via acylmethylation [87]. This protocol, inspired by the enzymatic processes of cytochrome P450, facilitates macrocycle formation through an innovative three-component coupling that directly employs unactivated C–H bonds and molecular oxygen (Figure 2a). This transformation is notably challenging to achieve using conventional synthetic methods. The remarkable efficiency and chemoselectivity of this reaction are largely attributed to the coordinated action of strategically positioned pyridine and ester functionalities, which facilitate intramolecular ring closure. Importantly, the resulting macrocycles prominently incorporate the biologically relevant aryl acetyl and aryl pyridine scaffolds, structural motifs frequently observed in various bioactive natural products.

The synthetic versatility of the rhodium(III)-catalyzed macrocyclization is further demonstrated by its extensive applicability to substrates containing ether, ester, and amide linkers, successfully yielding a diverse range of 17- to 27-membered macrocycles. Furthermore, the macrocyclic products obtained, especially those containing the α-aryl acetophenone moiety, function as significant intermediates for synthetic development into structurally intricate heterocyclic derivatives. Subsequent reactions facilitating the straightforward formation of C–N bonds effectively transform these macrocyclic intermediates into complex heterocyclic scaffolds, notably pyrido [2,1-a]isoindole-grafted derivatives (Figure 2b). The simplicity and efficacy of these transformations underscore the significant potential of this macrocyclization strategy to access complex and biologically intriguing architectures that would otherwise be challenging to synthesize through conventional methods. Furthermore, these biologically relevant macrocyclic derivatives exhibited significant antiviral activity, particularly against the H1N1 influenza virus. Phenotypic screening assays identified several compounds that effectively inhibited viral replication while demonstrating minimal cytotoxicity, thereby highlighting their therapeutic potential. Notably, compound **9**, a representative pyrido [2,1-a]isoindole-grafted macrocycle, demonstrated exceptional antiviral efficacy (EC₅₀ = 0.28 µM) along with a remarkably high selectivity index (>357), identifying it as a particularly promising candidate for antiviral drug development (Figure 2b).

Another cutting-edge macrocyclization strategy has been developed utilizing ligand-enabled palladum(II)-catalyzed sp³ C–H activation [88]. In 2024, Yang’s research group introduced a concise protocol for the direct cyclization of unactivated sp³ C–H bonds to synthesize macrocyclic sulfonamides, facilitating the one-step coupling of fragments (Figure 3a). This palladum-catalyzed approach permits direct intramolecular coupling, including alkylation, olefination, and arylation, at unactivated C–H bonds, resulting in the formation of 16- to 28-membered macrocyclic rings with privileged sulfonamide motifs, characterized by high structural diversity and biological significance, in a single synthetic step. By eliminating the requirement for prefunctionalized coupling partners, this strategy significantly simplifies macrocycle synthesis and allows for the incorporation of diverse building blocks that are not typically combined in natural pathways. A critical innovation was the use of monodentate ligands to accelerate the C–H activation and ring closure. Screening demonstrated that simple monodentate nitrogen ligands, such as substituted pyridines or quinolines, dramatically improved macrocyclization yields, whereas bidentate ligands (such as amino acid-derived ligands) gave poor results. The optimal ligand, 4-methylquinoline, facilitated efficient cyclization, producing the spiro-grafted macrocycles, while no macrocycle formation occurred in the absence of a ligand. It is hypothesized that an electron-rich heterocyclic ligand strongly coordinates with palladum(II), stabilizing the catalytic species and directing the formation of the macrocyclic bond. This ligand-driven enhancement significantly improved the method’s efficiency, enabling broad applicability across various linker types and ring sizes. Furthermore, diverse building blocks, including aromatic and aliphatic amino acids, as well as unnatural amino acids, dipeptides, tripeptides, and tetrapeptides, were seamlessly incorporated, achieving macrocycles with excellent yields and high diastereomeric ratios.

In an advanced development, an enantioselective variant of the sp³ C–H macrocyclization was established using a novel chiral silicon-substituted quinoline ligand in combination with a chiral dipeptide ligand (Figure 3b). The chiral catalytic system has effectively facilitated the synthesis of spiro-grafted macrocyclic sulfonamides featuring quaternary stereocenters, achieving enantioselectivities as high as 91.5:8.5 e.r. This advancement significantly broadens the array of synthetic methodologies available for the construction of pseudonatural macrocycles. The incorporation of asymmetric C–H activation into this catalytic system is particularly noteworthy, as it enables the production of enantioenriched macrocyclic structures that were previously difficult to synthesize.

In addition to synthetic innovation, the library of macrocyclic sulfonamides was explored for bioactivity, with a particular emphasis on mitochondrial sirtuin enzymes associated with neurodegeneration. An in silico docking screen of 1231 macrocycles against sirtuin 3 (SIRT3) identified several promising ligand candidates. Subsequent enzymatic assays confirmed that one macrocycle, compound **15**, significantly enhanced SIRT3’s deacetylase activity, indicating its role as a SIRT3 agonist (Figure 3c). Biophysical analyses further demonstrated that compound **15** binds directly to SIRT3 with high isoform selectivity. In cellular models of Parkinson’s disease (PD), a more soluble analogue (compound **13**) showed particular promise (Figure 3b). Compound **13** was found to engage SIRT3 within cells and, notably, to facilitate the clearance of toxic α-synuclein aggregates in neuronal cells, providing a protective effect against neuronal injury induced by neurotoxins. These protective effects were abrogated when SIRT3 was silenced, confirming that compound **13** acts through SIRT3 activation. Furthermore, in a PD mouse model, treatment with compound **13** significantly improved motor function and dose-dependently prevented the loss of dopamine (DA) neurons. This in vivo efficacy provides convincing proof-of-concept for SIRT3-targeted neuroprotection, underscoring the therapeutic potential of macrocyclic SIRT3 activators in neurodegenerative disease. Collectively, these modular biomimetic strategies represent a substantial advancement in macrocyclic synthetic methodologies, facilitating the efficient generation of structurally unique and pharmacologically relevant macrocycles with significant therapeutic potential.

## 3. DNA-Encoded Macrocycle Libraries (DELs)

DELs have emerged as a transformative methodology in the fields of drug discovery and chemical biology, providing a robust mechanism for the exploration of extensive chemical spaces [101,102,103,104]. This method combines the principles of combinatorial chemistry with the high-throughput capabilities of DNA sequencing. In DELs, each small molecule compound is covalently linked to a unique DNA sequence that serves as a barcode, allowing for the simultaneous synthesis and screening of millions to billions of compounds in a single mixture. The strength of DELs resides in their capacity to generate and screen vast collections of compounds, significantly surpassing the capabilities of conventional high-throughput screening techniques. This approach enables researchers to rapidly identify hit compounds against biological targets of interest. DNA tags facilitate the identification of active compounds and enable amplification, thereby allowing the detection of even rare binders. Furthermore, DELs can be designed to incorporate a diverse array of chemical frameworks and components, potentially leading to the discovery of novel chemical entities with unique properties or mechanisms of action.

Recent advances have significantly expanded DEL capabilities, notably through efficient solid-phase synthesis methods for generating extensive libraries of non-peptidic macrocycles. Kodadek et al. (2022) presented a robust solid-phase split-and-pool synthesis strategy to create diverse, thioether-linked macrocyclic libraries (Figure 4a) [89]. By strategically incorporating varied carboxylic acids and amines at multiple diversity points, their library achieved exceptional scaffold diversity, encompassing both 2.5-mer and 3.5-mer macrocyclic variants. Furthermore, their synthesis strategy intentionally reduced the presence of peptide-like amide N–H bonds, which are known to hinder cell permeability, thereby optimizing these macrocycles for improved intracellular targeting. To validate the library’s effectiveness, utilizing TentaGel resin simplified screening procedures, enabling straightforward identification of high-affinity and selective macrocycles using fluorescently labeled target proteins and fluorescence-activated cell sorting (FACS). Beads displaying macrocycles were incubated with fluorescently labeled streptavidin and sorted via FACS, allowing rapid decoding of binding hits through their unique DNA barcodes. Furthermore, detailed structure–activity relationship (SAR) studies were quickly performed using parallel solid-phase resynthesis of hit compounds, followed by subsequent FACS-based binding assays. Through these SAR analyses, crucial structural motifs influencing binding affinity were identified—most notably, the presence of an aminomethyl pyridine adjacent to a thiazole or oxazole backbone significantly enhanced affinity (Figure 4b). The macrocyclic compounds exhibited significantly enhanced binding affinities in comparison to their linear counterparts, highlighting the advantageous structural rigidity imparted by macrocyclization. Additional validation using fluorescence polarization assays confirmed that selected macrocycles exhibited high affinity and specificity toward streptavidin. Notably, investigations into cell permeability have demonstrated significant intracellular accessibility, thereby clearly indicating that intentional structural optimization remarkably enhances their therapeutic potential. This study collectively underscores the significant potential of DELs in facilitating macrocycle discovery by effectively integrating chemical synthesis with expedited biological evaluation. As a result, DEL technology represents a transformative approach for enhancing the identification and development of bioactive macrocycles aimed at addressing complex therapeutic targets.

## 4. Computational Design of Macrocycles

Recent advancements in computational chemistry and machine learning are increasingly complementing traditional laboratory-based methods in the pursuit of novel macrocyclic compounds [75,76,77,78,79]. A notable development in this area is the creation of deep learning models for macrocyclization. Liu et al. (2023) introduced a Transformer-based model named Macformer, which is capable of computationally generating macrocyclic analogues from specified linear molecules (Figure 1) [90]. The fundamental concept is to utilize AI to explore the extensive chemical space of macrocycles by virtually connecting different termini of acyclic molecules with a variety of linker fragments.

Macformer was trained to identify underlying relationships between linear precursors and their potential macrocyclic products, represented as Simplified Molecular Input Line Entry System (SMILES) strings, thereby enabling it to propose numerous chemically feasible cyclization outcomes. In practical application, when provided with an input structure, such as a known bioactive but flexible molecule, the model suggests a range of macrocyclic variants by incorporating linkers of varying lengths and compositions through which the molecule could cyclize. This methodology effectively automates the generation of macrocycle analogues, offering chemists a list of candidates with inherent conformational constraints. They demonstrated the utility of Macformer by applying it to the design of macrocyclic kinase inhibitors, specifically macrocyclized analogues of a JAK2 inhibitor. The generated macrocycles underwent filtering via molecular docking simulations, and several top-scoring candidates were subsequently synthesized and tested experimentally. Remarkably, this workflow resulted in the identification of potent JAK2 macrocyclic inhibitors (Figure 1), thereby validating the AI-guided design process. Beyond this specific case study, the deep learning approach addresses a broader challenge: systematically exploring the vast and discontinuous macrocycle space using in silico tools. Traditional medicinal chemistry might only attempt a limited number of cyclization variants due to synthetic constraints, whereas a model like Macformer can generate hundreds of possibilities and predict which are worth synthesizing. As training data and algorithms advance, such models may become increasingly proficient at suggesting macrocycles with not only strong binding potential but also favorable properties, such as predicting linkers that enhance cell permeability [86,87]. This integration of AI into macrocyclic design represents a new era where computational tools assist in navigating the complex design rules of macrocycles, complementing the experimental synthetic strategies previously described. It is noteworthy that computational methods are also employed to predict macrocycle conformations and properties, such as conformational sampling techniques to compute 3D descriptors, as discussed in a subsequent section on permeability. Collectively, these tools facilitate macrocycle discovery by directing synthetic efforts toward the most promising regions of chemical space.

In another noteworthy example, Barker et al. (2024) have made a significant advancement in the field of computational macrocycle discovery by developing a versatile computational method that systematically explores extensive and chemically diverse macrocyclic spaces (Figure 1) [91]. Unlike traditional computational approaches, which are predominantly limited to alpha-amino acids, their method facilitates the rapid identification and structural characterization of macrocycles derived from a comprehensive library of building blocks. This library includes alpha-, beta-, and gamma-amino acids, as well as other diverse backbone chemistries such as aminobenzoic acids, aminophenylacetic acids, and heterocyclic scaffolds. A key component of their strategy is a computationally efficient hashing algorithm, which enables the swift identification of conformationally feasible macrocycles by computationally pairing fragments whose backbone transformations permit ring closure. In particular, they computationally generated over 14.9 million chemically distinct macrocycles, utilizing more than 42,000 monomer combinations. This achievement highlights the method’s unparalleled capacity to explore this vast chemical space. Notably, from this extensive virtual library, they synthesized a selection of representative macrocycles, which were predicted to adopt single, energetically favorable conformations. Experimental validation through X-ray crystallography and NMR spectroscopy demonstrated remarkable agreement between the predicted and experimentally determined structures, thereby validating the reliability of their computational design strategy.

The practical utility and therapeutic potential of the computational approach were demonstrated through the successful design of macrocyclic inhibitors targeting therapeutically relevant proteins, including histone deacetylase 6 (HDAC6), the SARS-CoV-2 main protease (Mpro), and the MCL1–Bak protein–protein interaction (PPI) (Figure 1). Notably, several macrocycles exhibited potent and selective inhibitory activities, thereby validating the computational predictions and underscoring the approach’s potential to facilitate the development of novel drug candidates. Significantly, the computationally designed HDAC6 inhibitors displayed notably increased potency and selectivity compared to simpler linear analogues, illustrating the value of conformationally constrained macrocycles in enhancing therapeutic specificity. Furthermore, extensive permeability assays, specifically the parallel artificial membrane permeability assay (PAMPA), have confirmed that numerous computationally designed macrocycles exhibit desirable drug-like characteristics, including favorable membrane permeability. This finding addresses typical challenges in the field of macrocycle drug discovery.

Collectively, these findings highlight the transformative impact of computational methodologies in the discovery of macrocycles. By systematically exploring chemically diverse and structurally complex macrocycle spaces, these computational approaches complement experimental techniques, guiding synthetic efforts toward regions of macrocycle chemical space that are highly promising yet traditionally challenging to access experimentally. This integrated computational–experimental workflow thus represents a significant advancement in the rational and systematic development of bioactive macrocyclic compounds. Looking ahead, the integration of generative models may facilitate the automated design of macrocycles with multiple optimized properties. Furthermore, recent advances in conformer prediction hold promise for better capturing the chameleonic behavior and folding-driven permeability of macrocycles. These AI-guided strategies may soon enable holistic optimization of structure, bioactivity, and ADME in a single computational pipeline.

## 5. Cell Permeability and Bioavailability of Macrocycles

A critical aspect of evaluating any macrocyclic drug candidate involves the assessment of its absorption, distribution, metabolism, and excretion (ADME) properties, with particular emphasis on cell permeability and oral bioavailability [37,105]. Macrocycles often deviate from Lipinski’s “rule of 5” guidelines, as they are typically larger and more polar than conventional small molecules [31]. Nonetheless, many macrocycles can penetrate cell membranes and achieve oral absorption, defying classical expectations. This apparent paradox is explained by the unique ability of macrocycles to adopt specific folded conformations that conceal their polar groups, a phenomenon known as “chameleonic” behavior [106,107,108]. By internally sequestering hydrogen bond donors and acceptors and minimizing polar surface area in a given conformation, a macrocycle can present a more lipophilic face to the membrane, thereby facilitating diffusion. For example, cyclosporine A, a cyclic undecapeptide (~1200 Da), is orally bioavailable and can traverse cell membranes due to a conformation in which numerous amide NH and C=O groups form intramolecular hydrogen bonds, significantly reducing the polar surface exposed to the solvent (Figure 2a) [106]. While cyclosporine A exemplifies how favorable macrocyclization can be for permeability and bioavailability, the impact of cyclization varies significantly across different compounds. Such molecular preorganization can influence not only permeability but also affinity to biological targets and thus significantly enhance bioactivity. In the field of medicinal chemistry, it has been observed that the macrocyclization of a linear molecule can, at times, enhance its bioactivity. A particularly notable example involves the stabilization of the bioactive conformation through macrocyclization, which resulted in an approximately 55,000-fold increase in potency compared to its linear analogue, as evidenced by a shift in IC₅₀ from micromolar to nanomolar concentrations (Figure 2b) [109]. This extreme case, involving a farnesyltransferase inhibitor, illustrates how the preorganization afforded by a macrocyclic structure can eliminate the entropy cost associated with binding, thereby leading to significant enhancements in potency. Nevertheless, the impact of macrocyclization on permeability is not universally beneficial; some studies have reported only minor improvements or even decreases, contingent upon the scaffold [110,111,112]. This variability underscores the complexity of permeability and emphasizes the necessity for strategy-specific evaluations. As summarized in Table 3, various structural modifications, including functional group alterations, cyclization, backbone constitution, and stereoinversion, can exert significantly different effects on membrane permeability, depending on the macrocycle’s conformational behavior and polarity. The outcome is dependent on the specific structural characteristics, as macrocyclization may inadvertently result in an excessively polar ring or constrain the molecule into a conformation that is incompatible with membrane permeability. Therefore, predicting macrocycle permeability in advance is challenging and requires a thorough analysis of molecular conformations.

To elucidate and predict the permeability of macrocyclic compounds, Kihlberg et al. (2021) employed 3D conformation-dependent descriptors [92]. Traditional 2D metrics, such as calculated lipophilicity (cLogP) or topological polar surface area (TPSA), frequently fail to correlate with the permeability of large, flexible molecules, as these descriptors do not account for a macrocycle’s capacity to shield polarity through folding. Conversely, descriptors derived from an ensemble of low-energy 3D conformers have demonstrated greater informativeness. For example, the study examined pairs of diastereomeric macrocycles that exhibited significantly different human colon adenocarcinoma (Caco-2) cell permeabilities despite having identical formula (Figure 3a). It was observed that descriptors such as the solvent-accessible 3D polar surface area (SA 3D-PSA) and the radius of gyration (*R*_gyr_, a measure of molecular compactness) correlated well with the permeability differences between stereoisomers. These descriptors were obtained by generating representative conformational ensembles for each macrocycle, utilizing molecular dynamics and NMR data to ensure relevance, and subsequently computing the properties on those 3D structures. The results consistently indicated that macrocycles adopting more compact, internally hydrogen-bonded conformations—characterized by reduced radii of gyration and smaller 3D polar surface areas—displayed higher cell permeability. Notably, traditional 2D parameters failed to distinguish permeability differences between isomers, whereas these 3D descriptors accurately ranked their permeabilities. This highlights the critical importance of considering a macrocycle’s ensemble of conformations; two isomers may have identical polar groups and lipophilicity in theory, yet structural folding can significantly alter their permeability. By sampling conformers through computational methods (such as Monte Carlo or molecular dynamics simulations) and evaluating their polar surface area, hydrogen bonding patterns and molecular compactness, it becomes possible to more reliably predict membrane permeability in macrocyclic drug candidates (Figure 3b).

Encouraged by the compelling results in predicting cell permeability based on molecular structure and conformational investigations, computational explorations present a promising opportunity to enhance pharmacokinetic development during macrocyclic drug discovery. Still, the macrocycle-based drug discovery needs to optimize its unpredictable ADME profiles, focusing on pharmacokinetic properties through structure–activity relationship studies [116], including solubility, stability, and permeability [37,117]. By integrating the computational designs related to structural properties discussed in Chapter 4, we gain valuable insights that enhance our understanding of pharmacokinetics in the context of macrocyclic drug discovery.

## 6. Conclusions

In conclusion, recent advancements in synthetic and computational methodologies have observably progressed the field of macrocycle drug discovery, addressing longstanding challenges related to their complex synthesis, structural optimization, and pharmacokinetic properties. Innovative approaches, such as modular biomimetic assemblies and DELs, have effectively expanded accessible chemical spaces, facilitating the creation of structurally diverse and biologically potent macrocycles. Concurrently, computational strategies utilizing advanced algorithms and AI have streamlined macrocycle design by accurately predicting structures and properties critical to therapeutic efficacy. Furthermore, an enhanced understanding of macrocycles’ conformational adaptability, particularly their ability to internally shield polar groups to improve cell permeability, has significantly informed rational design approaches. Through the synergistic integration of these synthetic and computational methods, promising macrocyclic candidates have emerged, demonstrating substantial therapeutic potential across various biomedical applications. Continued development and refinement of these integrated strategies are likely to lead to further breakthroughs, firmly establishing macrocycles as versatile and effective therapeutic agents capable of addressing previously undruggable biological targets.

However, the advancement of macrocyclic compounds as potential drug candidates is impeded by challenges associated with production costs, synthetic scalability, and potential toxicity. The elevated production costs are primarily attributable to the intricate synthetic pathways and the requirement for specialized reagents, which lead to higher manufacturing expenses compared to traditional small-molecule drugs [117]. Furthermore, the transition from laboratory-scale synthesis to industrial-scale production presents technical challenges that can further increase costs. From a safety perspective, certain macrocyclic scaffolds have shown safety concerns; for instance, polymyxins have been associated with nephrotoxicity [118], while daptomycin has been linked to drug reaction with eosinophilia and systemic symptoms (DRESS) syndrome [119], underscoring the importance of early toxicity assessments during drug development. To render the development of macrocyclic drugs more clinically and economically viable, it is crucial to advance synthetic methodologies, incorporate predictive safety evaluations, and refine reaction conditions. Addressing these challenges through innovative design and cost-effective synthetic and toxicological strategies is essential for the successful clinical translation of macrocyclic drug candidates.

## Data Availability

Not applicable.

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
