# Peer review of "Exploring Macrocyclic Chemical Space: Strategies and Technologies for Drug Discovery"

_pharmaceuticals, 2025, doi:10.3390/ph18050617_

Round 1

Reviewer 1 Report

Comments and Suggestions for Authors

The review article by Taegwan Kim et al entitled ‘Exploring Macrocyclic Chemical Space: Strategies and Technologies for Drug Discovery’ describes the progress in the field of macrocycle drug discovery contributed by the advancements in synthetic and computational methodologies. Also, with the recent advancements in AI and computational capabilities aid to better design structurally diverse and biologically active macrolytes.  The article is extensive and well written with a major emphasis on the chemical structure and properties. The figures are well presented with macrolyte examples, synthesis, and cyclization potentials. The article can be further improved by adding more on the functional or mechanism of action (ligand binding) aspects in the introduction or in the discussion section to hold more interests to the readers. Minor suggestions include,

  1. The therapeutic aspects or progress of the macrocycles can be discussed or included as a table. Examples include recent advances in macrocycle-drugs for cancer treatment and infectious diseases.
  2. A table with the list of examples of targets for approved macrolytes can be included.
  3. Emphasis on the overall cost or toxicity status of macrolye-based drugs is missing/minimally discussed in the review.

Author Response

Please see the attached file for our point-by-point response to the reviewers' comments.

Reviewer 2 Report

Comments and Suggestions for Authors

Present manuscript entitled ‘Exploring Macrocyclic Chemical Space: Strategies and Technologies for Drug Discovery’ presents a timely and comprehensive review of recent synthetic and computational strategies that have expanded the macrocyclic chemical space in drug discovery. It effectively highlights both the challenges and advances in macrocycle development, particularly emphasizing innovative synthetic methodologies, DNA-encoded libraries (DELs), and AI-driven design. The integration of case studies involving antiviral and neuroprotective macrocycles strengthens its practical relevance. The review is well-organized, clearly written, and supported by extensive and appropriate citations.

I feel the work deserves publication in Pharmaceuticals. However, the following minor revisions should be incorporated before consideration for publication:

Strengths:

Thorough Coverage: The review spans recent developments in both synthetic and computational macrocycle research over the past five years, including high-impact strategies like modular biomimetic assembly and DELs.

Integration of Case Studies: Practical applications are well illustrated through examples of antiviral (H1N1) and neuroprotective (SIRT3 agonists for PD) macrocycles.

Computational Insights: The discussion of AI and deep-learning–driven macrocyclization provides cutting-edge insight into the future of drug discovery workflows.

Clarity and Flow: The manuscript is logically structured and easy to follow, making it accessible to a broad scientific audience.

  • Suggestions for Improvement:

Figures and Schemes: Ensure all figures (e.g., Scheme 2, Figure 2) are of high resolution and clearly labeled. A graphical summary or ‘key challenges & future outlook’ schematic could enhance visual impact.

Depth of Challenges: While challenges in macrocycle synthesis are addressed, a more detailed discussion on regulatory, pharmacokinetic limitations, or scale-up issues could provide a broader perspective. Include recent examples.

AI and Machine Learning: The Macformer and fragment-hashing approaches are described well, but future directions for AI integration (e.g., generative models, conformer prediction) could be more elaborated.

Permeability Section: Including a table summarizing examples with different permeability outcomes (positive, neutral, negative) would offer a clearer comparison.

This manuscript is a valuable contribution to medicinal chemistry and macrocyclic drug discovery. With a few refinements to clarity and visual representation, it will serve as a strong reference for researchers across disciplines.

Author Response

(The authors gave the same response as above.)
